# A Time-Efficient Co-Operative Path Planning Model Combined with Task Assignment for Multi-Agent Systems

**Sumana Biswas** *[ID], **Sreenatha G. Anavatti** and **Matthew A. Garratt**[ID]

Defence Force Academy, University of New South Wales, 2052 Canberra, Australia;
S.Anavatti@adfa.edu.au (S.G.A.); M.Garratt@adfa.edu.au (M.A.G.)
* Correspondence: Sumana.Biswas@student.adfa.edu.au

**Abstract:** Dealing with uncertainties along with high-efficiency planning for task assignment problem is still challenging, especially for multi-agent systems. In this paper, two frameworks—Compromise View model and the Nearest-Neighbour Search model—are analyzed and compared for co-operative path planning combined with task assignment of a multi-agent system in dynamic environments. Both frameworks are capable of dynamically controlling a number of autonomous agents to accomplish multiple tasks at different locations. Furthermore, these two models are capable of dealing with dynamically changing environments. In both approaches, the Particle Swarm Optimization-based method is applied for path planning. The path planning approach combined with the obstacle avoidance strategy is integrated with the task assignment problem. In one framework, the Compromise View model is used for completing the tasks and a combination of clustering method with the Nearest-Neighbour Search model is used to assign tasks to the other framework. The frameworks are compared in terms of computational time and the resulting path length. Results indicate that the Nearest-Neighbour Search model is much faster than the Compromise View model. However, the Nearest-Neighbour Search model generates longer paths to accomplish the mission. By following the Nearest-Neighbour Search approach, agents can successfully accomplish their mission, even under uncertainties such as malfunction of individual agents. The Nearest-Neighbour Search framework is highly effective due to its reactive structure. As per requirements, to save time, after completing its own tasks, one agent can complete the remaining tasks of other agents. The simulation results show that the Nearest-Neighbour Search model is an effective and robust way of solving co-operative path planning combined with task assignment problems.

**Keywords:** multi-agent system; task assignment; path planning; dynamically changing environments

## 1. Introduction

There is an increasing trend towards using autonomous systems cross a wide range of real-world applications where human presence is deemed unnecessary or dangerous. Nowadays, Unmanned Vehicles (UVs) are deployed for various missions including accompanying troops in the battlefield, security enforcement, surveillance, etc. As the mission becomes more complicated, they may need to accomplish multiple tasks in dynamic environments. Completion of the successful mission depends not only on the smooth exploration of the workspace but also on the accomplishment of a set of tasks spread over an extended region [1].

It is always difficult for a single agent system to perform a wide range of missions, due to its limited adaptation, non-flexibility and low reliability characteristics. On the other hand, multi-agent systems are very popular for their robustness and self-adaptation capacity [2]. They are getting more

and more emphasis and research focus [3,4]. Productivity enhancing and vehicle safety are the two main factors that should be satisfied in all stages of multi-agent planning [5].

Planning a safe and efficient path for each UV is still one of the challenges for mission planning. Obstacle avoidance, as well as collision avoidance among agents, plays an important role in the context of managing multiple agents [6]. On the way to the selected locations, all UVs should be able to avoid obstacles by replanning their path dynamically.

Sharing the burden among multiple agents can improve the effectiveness of a mission [7]. During planning, the paths for all UVs in the scenario are determined by assigning certain number of tasks to each individual UV. In such scenarios, UVs are required to move in the search space through the designated locations associated with the given task [7]. Complications arise during decision making like, "who will go to which locations in what order" [7]. Furthermore, when UVs move together, there is a risk of colliding with each other. Efficient coordination among multiple agents is the key requirement for successful mission planning [8]. Hence, an effective cooperative path planning with task assignment should be needed to make the mission successful.

It is not an easy task to measure the performance of mission planning algorithms and compare one with the other. Moreover, planning approaches are different in structure and their implementations depends on the representations of the environments [9]. The performance of a framework depends on several factors that are related to the solution of the tasks for which the framework is being used [9]. In a mission, different types of uncertainty, such as break down of UVs, and task switching (to reduce the mission competition time) exist. An efficient framework is needed to address these challenges.

This work addresses the comparison between two different frameworks proposed by Sumana et al. [10,11]. The extension of the work in [10] is also developed in this paper. Both frameworks are based on the Particle Swarm Optimization (PSO) method and are used for efficient cooperative path planning for a multi-agent system. PSO offers a high-quality solution and fast computation for solving different complex problems. In terms of path planning, it leverages the advantages of the heuristic and random search strategies used in PSO. Both frameworks can efficiently work in complex environments. Simultaneous replanning is applied to replan a new path by avoiding both static and dynamic obstacles. Both approaches are capable of avoiding collisions among agents. Here, the co-operative manner of path-planning with integrated collision avoidance is combined with the task assignment problem. It is assumed that information collected by sensors is shared among all agents. Thus, each agent has information about the other agents' positions. Multiple agents can automatically arrange the total given task and dynamically adjust their motion. Sumana et al. [11] proposed a Compromise View (CV) model as the task allocation approach. A Nearest-Neighbour Search (NNS)-based task allocation model is proposed in [10].

The contribution of this paper is to analyze and make comparative assessment of these two models. Moreover, this paper addresses whether the NNS approach is capable of dealing with uncertainties such as break down of any agents. In addition, this paper addresses the complicated situation where the number of agents is less than the number of tasks. To increase the efficiency of the total planning, agents also have the capability to a switch few of their assigned tasks to another agent who has already finished its tasks. Simulation results validate the effectiveness of the NSS model over the CV model.

The rest of the paper is organized as follows: Section 2 describes the literature review related to this topic. Section 3 states the problem under consideration more precisely. Section 4 defines the frameworks and describes the methodology of the total planning. Simulation results and discussions are presented in Section 5. The concluding remarks are given in Section 6.

## 2. Related Work

Since the 1960s, much research has been conducted on path planning [12–16]. Generally, there are two approaches of path planning. One is centralized and the other one is decentralized. In a centralized approach, the planner computes the path in a combined configuration space and treats the agents as a single combined agent [17]. Clark et al. [18] used dynamic networks that coordinate centralized

planning for effective motion planning of multiple mobile agents. On the other hand, the decentralized approach computes a path for each agent independently [17]. The decentralized approach is a generalization of behavior-based control of agents and they are faster than the centralized approaches. Cascone et al. [19] used a decentralized approach in multiple agent planning.

Recently, swarm agent systems based on the concept of swarm intelligence have become a very active research area for multi-agent systems. Particle swarm optimization (PSO) [20] based on the concepts of swarm intelligence, has become an excellent optimization tool. It generates a high-quality solution with low computational time [21]. It has few parameters and it can converge in a very fast manner [22]. Compared with other methods, such as genetic algorithms [23], PSO gets better results in a faster and cheaper way. Leveraging the qualities of rapid search and easier realization, PSO attempts to solve the drawbacks of conventional/existing methods [24]. Sumana et al. [25] used PSO based algorithm to solve path planning problems.

In the context of multi-agent systems, path planning problems become more complicated when solving for obstacle avoidance and collision avoidance among agents [6]. Zhang and Zhao [26] proposed an A*-Dijkstra-Integrated algorithm for multiple agents without any collision. Peng et al. [27] proposed an improved D* Lite algorithm with a fast replanning technique. However, in most of the research, the agents are considered as point objects and only consider the static environments. Moreover, most of the planner implies a very limited class of mission execution [28], such as single start to single goal [29,30].

In many mission planning problems, a group of agents need to execute a series of tasks on their way to the destination. There has been a considerable amount of work carried out on task assignment problems. An optimal assignment algorithm is used by Kwok et al. [31] to complete multiple targets in a multi-agent scenario. Akkiraju et al. [32] proposed an effective agent-based solution approach. To achieve cooperation between autonomous agents, Noreils [33] proposed an effective planning method. However, the focus of those algorithms is only on task allocation problems without combining it with path planning. Path planning combined with task assignment is a NP-complete problem [1]. Berhault el al. [34] utilize the advantage of combinatorial actions to coordinate a team of mobile robots to visit a number of given targets in partially unknown terrain.

Mission planning combined with task assignment becomes more complicated when it works in the dynamic environments. Multi-agent path planning in dynamic environments is proposed by Jan et al. [35]. An evolutionary approach to a cooperative mobile robotic system is implemented by Yu et al. [2]. Maddula et al. [1] provide an excellent overview of multi-target assignment as well as path planning for a group of agents. A negotiation-based algorithm is applied by Moon et al. [7,36] for assigning tasks to agents. To find the shortest path in a dynamic environment, they combined this algorithm with the A* search algorithm [37]. Zhu and Yang [38] applied a neural-network-based approach to solve dynamic task assignment of a multirobot system. However, most of the research work does not consider the size constraint of the agents. Moreover, to optimize the computational cost, most of the existing approaches do not have an efficient replanning capacity.

Sumana et al. [10,11] proposed two different frameworks of a multi-agent system for cooperative task assignment combined with path planning. This work is the extension of the previous work, which compares both frameworks from [10,11]. Furthermore, in this paper, the better framework is extended to deal with the breakdown problems in addition to switching workload to make the mission time efficient.

## 3. Problem Description

Consider a given number of agents, $M = \{M_1, M_2, \ldots \ldots M_i\}$ and a task set, $T = \{T_1, T_2, \ldots \ldots T_i\}$ that needs to be completed by the agents. The target locations are randomly distributed in the planned area. The position of the target location is denoted by $(X_{iT}, Y_{iT})$, $i = 1, 2, \ldots T$. The position of agents 'm' is given by $(X_{iM}, Y_{iM})$, $i = 1, 2, \ldots M$. Here 'M' < 'T'. Without loss of generality, the task is to visit a number of target locations. Each agent is required to visit at least one location.

The challenge is assigning the number of tasks to multiple vehicles on its way to the destination. $O = \{O_1, O_2, \ldots \ldots O_i\}$ is the set of static and dynamic obstacles that remain in the scene at a given time instant. In this problem, let each agent initially be located at some initial point 'S', and 'F' be the final goal location. The flexibility of the planning is that it is not necessary that the goal point of each agent will be the same; they can be different as well. If 'P' is the path, the assigned path to each agent is given by: $Pi = \{(X_{iM}(s), Y_{iM}(s)), (X_{i1T}, Y_{i1T}), \ldots (X_{iTT}, Y_{iTT}), (X_{iM}(f), Y_{iM}(f))\}$. On its path, agents need to avoid any types of obstacles. For this stage, the total solution concept is adopted from Sumana et al. [10,11]. The sequences of the problem and the rules are given below.

- Divide the total set of tasks into subset of tasks.
- Assign subset of tasks to individual agents.
- Agents must ensure collision free path while following their paths. Therefore, the paths of all agents solved by the proposed framework should be admissible.
- Agents must find the shortest path.
- Tackle unwanted situations such as break down of any agent.
- If needed task switching can be applied.

Here some assumptions are considered to simplify the representation. In the real world, autonomous agents and obstacles are of heterogeneous shapes, so a circle is considered around the agents and obstacles. Here, the agents are modelled as a circle with $R_1$ radius. It is also considered that the agent moves at a constant speed without any restriction on turning.

A safety gap is considered by extending the radius of the enclosed circle for both agents and obstacles. One agent considers the other agents as dynamic obstacles and apply the same collision avoidance strategy.

It is also considered that each agent has information about other agents and about surrounding environments.

## 4. Solution Approaches

The frameworks are composed of three layers. They are Path planning, Task allocation and Collision avoidance. Figures 1 and 2 present the frameworks for the cooperative mission planning. Figure 1 shows the CV model and Figure 2 describe the NNS model.

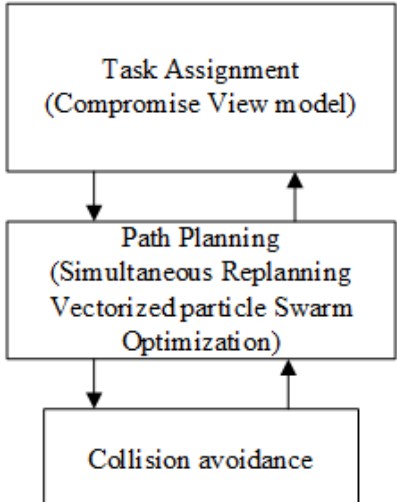

**Figure 1.** Flow diagram of framework based on the Compromise View (CV) model.

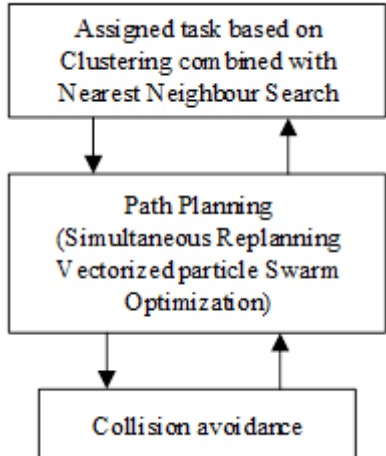

**Figure 2.** Flow diagram of the Nearest-Neighbour Search (NNS) model.

In a multi-agent system, the integration of the task assignment layer with the path planning layer is the most significant part for architecting a mission planner [7]. In the CV model, the task assignment layer gets the cost information from the path-planning layer, where the cost is the selecting criterion for task assignment. On the other hand, in the NNS model, the task assignment layer collects information about the target locations. Based on this information, a cluster of tasks is formed. A combination of the clustering method with the Nearest-Neighbour Search model is used to assign tasks to each agent. Iterations continue within these two layers, until a feasible solution for task assignment is found.

The objective of the path planning layer and the collision avoidance layer is to do a similar task of finding a collision-free path. The only difference is that the collision avoidance layer replans a path to avoid collisions. Hence, these two layers work in a complementary manner to generate a collision-free path efficiently.

### 4.1. Path Planning Layer

In 1995, Kennedy and Eberhart introduced PSO [20]. PSO works with a swarm population of particles that represents a possible solution. It adjusts its exploration and exploitation capacity of search according to its own and its companion's flying experience. The personal best position is the best solution that is found by the particle in the course of flight. The best position of the whole flock is the global best solution [38]. Particles update their positions by updating their velocities. The updating process runs until an optimum solution is reached. The particles update their velocities and positions by the following equations:

$$v_i(t + 1) = \omega v_i(t) + c_1 r_1 (x_i^{pbest}(t) - x_i(t)) + c_2 r_2 (x_i^{gbest}(t) - x_i(t)) \tag{1}$$

$$x_i(t + 1) = x_i(t) + v_i(t + 1) \tag{2}$$

where, $c_1$ = cognition parameter, $c_2$ = social parameter, $\omega$ = inertia weight factor, and $r_1$, $r_2$ are independent random numbers uniformly distributed in [0, 1]. $x^{pbest}$ is the personal best position achieved by the particle and $x^{gbest}$ is the global best position achieved by the entire swarm of particles.

By updating their velocities and positions, a new generation forms and particles move forward for a better solution. Ultimately, all particles converge onto the optimal solution. Practically, the fitness function of the optimization problem assesses the solution to which the particle is good or bad [39].

In this work, the Simultaneous Replanning Vectorized PSO (SRVPSO) algorithm from [29] is applied. Algorithm 1 describe the SRVPSO algorithm.

---

**Algorithm 1: Simultaneous Replanning Vectorized**
Particle Swarm Optimization for Path Planning

---

**Input:** Number of population (swarm_size), maximum
number of iterations (max_it), Initial positions ($x_s$),
Goal position ($x_F$), Initial velocity (v), $c_1$, $c_2$, $\omega$.
**Output:** Way-points

---

1.  **while** number of iteration $\leq$ max_it do
2.   **for** each particle
3.    evaluate fitness function
4.     **if** fitness (x) > fitness (pbest); pbest = personal best
5.     pbest = x
6.     **if** fitness (x) > fitness (gbest); gbest = global best
7.     gbest = x
8.     update particle's position and velocity according to Equations (2) and (3)
9.     **for** each particles position check the risk of collisions
10.    calculate the distance between obstacles and particles, $D_p{}^{obs}$
11.     **if** $D_p{}^{obs} > 0$; no collision
12.     **else** $D_p{}^{obs} \leq 0$; collision is likely to place
13.     Replan the path to avoid obstacles.
14.     **end if**
15.     **end for**
16.     **end if**
17.   **end for**
18.   gbest = parameters of best solution
19. **end while**

---

*4.2. Task Allocation Layer*

Task allocation is the most significant part of the aforementioned frameworks. Algorithm 2 and Algorithm 3 describe the task assignment layers, used by Sumana et al. [10,11], and their comparisons are given below.

---

**Algorithm 2: Algorithm for Task Assignment (k-NNS model)**

---

**Input:** All target locations ($T_{i,}$), All agents ($M_i$)
**Output:** Assigned target locations $T_j \in T_{i,}$ for $j^{th}$ agent $\in M_i$

---

1. Divide all target locations Ti into M number of clusters C using k-means clustering
2. Calculate the centroid of the clusters
3. **for** all M $\in M_i$
4.  Calculate the nearest cluster $C_{nearest}$ considering closest distance between the centroid of the clusters and the position of the agents
5.  Assign $T_j$ targets inside the cluster for $j^{th}$ agent
6. **end for**
7. **return** $T_j$

---

---

**Algorithm 3: Algorithm for Task Assignment (CV model)**

---

**Input:** All target locations ($T_{i,}$), All vehicles ($M_i$)
**Output:** Assigned target locations $T_j \in T_{i,}$ for $j^{th}$ agent $\in M_i$

---

1. Calculate the associated cost with all the target locations
2. All agent chose its first target with the lowest cost.
3. **for** all $M \in M_i$
4.   Conflict arise, when more agents select the same target location
5.   Resolve conflict by applying CV model; conflicting agents compare their costs and the agent with lowest cost is assigned to cover the target location.
6.   Repeat step 1 to 5 until all the agents are assigned a target location
7. **end for**
8. **return** $T_j$

---

### 4.2.1. Compromise View (CV) Model

The CV model by Sumana et al. [11] is used to select targets and assign them to agents. This approach is a heuristic method and it keeps updating until all the target locations are visited by the agents. The CV model is performed in the following manner.

(I)　Until all the Target Locations are Visited by the Agents

- Each agent calculates the cost associated with all the target locations and chooses its first target with the lowest cost. Let, $\left(\text{Current\_x}_{im}, \text{Current\_y}_{im}\right)$ is the current coordinate of 'M' agents and $(x_{iT}, y_{iT})$ is the coordinate of 'T' targets. The cost is defined as follows:

$$\text{PD} = \sqrt{\left(\text{Current\_x}_{iM} - x_{iT}\right)^2 + \left(\text{Current\_y}_{iM} - y_{iT}\right)^2} \tag{3}$$

Here, PD is the path distance from the current position of the agent to the target location. All agents get information about the choices of other agents. A conflict arises when more agents select the same target location. The following actions are taken to resolve the conflict:

- This conflict is resolved by the compromising view of the conflicting agents, i.e., conflicting agents compare their costs and the decision goes in favour of the agent with the lowest cost.
- Iteration is carried over all the agents and targets until all conflicts are resolved or all the agents are assigned a task in the current cycle.

(II)　Repetition of step I

When each agent is assigned with a task, the path planning layer starts to plan a feasible path.

### 4.2.2. Nearest-Neighbour Search (NNS) Model

This NNS model is used by Sumana et al. [10] for task assignment purposes. Here, k-means Clustering is used to divide all the target locations into a number of clusters. The number of clusters is the same as the number of agents. At the next step, the NNS model is applied to assign the cluster of tasks to individual agents. Each agent calculates the distance between its own position and the centroid of the cluster, then the cluster of tasks that are near to agents is assigned their target locations. If the same cluster is assigned to more than one agent, a conflict may arise as to who will cover the assigned cluster of tasks. A cluster refined technique is applied to resolve the conflict. By the refining process, once the cluster of target locations assigned to its nearest agent, it will remove from the clustering list. Thus, this cluster will no longer be assigned to other agents. Finally, when each agent is assigned with a cluster, the algorithm goes to its next stage of path planning by covering all of its assigned tasks.

### 4.2.3. Comparison between CV Model and NNS Model

In the CV model, tasks are assigned to agents in a sequential manner. That is the main drawback of this method. At each iteration, the distance between agents' current position and the entire target locations are needed to be calculated that is time consuming. Hence, the computational time of this method is high.

On the other hand, in the NNS model, at the very beginning of the method, all targets are clustered according to the number of agents. Here a group of targets is assigned at a time to individual agents. This procedure is less time consuming.

Table 1 shows the comparison between these two models. From the Table 1, it is found that the computational time of CV model is higher than the NNS model. Hence, the NNS model is more computationally time-efficient than the CV model. However, it takes a longer path than the CV model.

**Table 1.** Comparison between CV and NNS Model.

| Initial Position of 2 Agents | Goal Position of 2 Agents | No. of Targets | No. of Static Obstacles | No. of Dynamic Obstacles | Computational Time (s) | | Pathlength (m) Agent-1 | | Pathlength (m) Agent-2 | |
|---|---|---|---|---|---|---|---|---|---|---|
| | | | | | CV | NNS | CV | NNS | CV | NNS |
| **(0, 0)** | **(100, 100)** | **10** | **4** | 0 | 5.62 | 1.75 | 229.98 | 309.31 | 167.41 | 301.72 |
| (0, 0) | (105, 105) | 14 | 3 | 2 | 5.54 | 1.69 | 250.81 | 297.61 | 171.92 | 283.84 |
| (0, 0) | (0, 0) | 10 | 4 | 0 | 5.61 | 2.53 | 202.82 | 375.21 | 162.21 | 346.47 |
| (0, 0) | (80, 77) and (60, 93) | 14 | 3 | 2 | 5.60 | 1.89 | 199.68 | 296.18 | 136.83 | 283.15 |

### 4.3. Collision Avoidance Layer

In multi agent planning, agents have to work simultaneously on the same work space. To find the shortest trajectory that avoids collision with obstacles and collision with other agents is still a significant challenge for the planner [40]. Agents need to take necessary action as soon as they detect the position and movements of obstacles. In this case, coordination and cooperation among agents are the key concern of safety planning. Furthermore, they require replanning their path to avoid the risk of collisions. Here, one agent considers the others as dynamic obstacles and applies the same obstacle avoidance strategy to avoid collisions among them.

The collision avoidance strategy is integrated with path planning layer. It works as a reactive planner. A two-stage collision avoidance strategy is followed in this SRVPSO algorithm. The strategy is explained as follows:

Stage I: each particle calculates the cost (distance) from its current location to the obstacles position. Let $D_p^{obs}$ be the distance between each obstacle and the particles, shown in Figure 3.

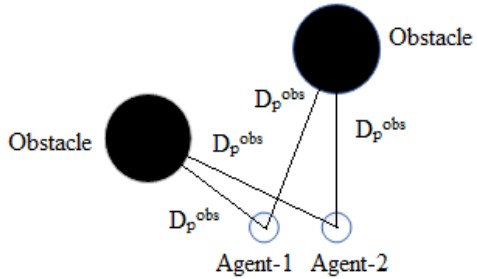

**Figure 3.** Distance between obstacles and particles.

Rules for collisions occurrence:

- If $D_p^{obs} > 0$, no collision, move to next step.
- If $D_p^{obs} \leq 0$, collision is likely to take place, then agents have to replan their path according.

Here, the negative value of the cost means that the trajectory of the particles is within the collision zone. The fitness value of the particles within the risk zone is manipulated by a higher positive value. This optimization problem is a cost minimization problem and the particles with higher cost are considered to have the worse fitness function. Thus, the trapped particles in the collision zone are ignored and the path within the zone is infeasible.

Stage II: To overcome the unbalanced scenario in the swarm population, the trapped particles are recovered. The parameters of PSO algorithm recovering procedure is carried out by increasing the values of the PSO parameters that increase the velocity of these particles. Thus, they can be recovered from the collision zone.

By applying this strategy, multi-agents can easily avoid any type of collision and replan the path that is admissible.

## 5. Simulation Results and Discussions

The SRVPSO algorithm is coded in MATLAB R2016b and tested on a Windows computer with Intel(R) Core(TM) i7-4770 CPU @3.40 GHz and 16.0 GB of RAM.

The parameters of PSO used in the tests have been chosen by running several tests with different combinations. The preferred combinations of parameters are as follows:

- $\omega = 1.0$, $c_1 = 0.5$, $c_2 = 2.0$
- Population size of swarms (agents) = 25 and
- Maximum iterations = 50

Parameters considered for collision avoidance strategy are as follows:

- $\omega = 1.0$ and $c_1 = 3.0$, $c_2 = 4.0$

These increased parameter values help the trapped particles to recover.

To evaluate the performance of the total framework, several tests on different cases are simulated. A two dimensional $120 \times 120$, $100 \times 110$, and $300 \times 300$ square unit environment is considered as the search space. Different types of working environments with different types of obstacles with several numbers of task assignments are considered. For each configuration, all the simulations run several times.

The solid circles represent the static obstacles and the hollow circles represent the dynamic obstacles (Figures 4–14). The hollow circles' sequence represents the trajectory of dynamic obstacles. In Figures 4–7, the agents are represented as small circles. The rest of the figures show the tracked path of the agents during path planning covering the target locations.

The dynamic obstacles can have the constant speed, or they can be randomly moving objects. The diamond shapes represent the target locations that the agents need to visit. Different types of scenarios for mission planning are considered below.

### 5.1. Static and Dynamic Environments with Various Number of Agents and Targets

Figures 4 and 5 show the path planning and task assignment problem for two agents in the dynamic environments.

In Figure 3, the number of assigned targets is fifteen whereas in Figure 4 the number of assigned targets is sixteen. In Figure 3 both agents come in the same goal position, and in Figure 4 agents reached in different goal locations. Here, the task assignment problem is solved by using the CV model.

Figure 6 shows the path planning and task assignment problem of three agents in the static environment. Here, the number of target locations is eight. Figure 7 shows the path planning of three agents combined with task assignment problem in a complex environment. In Figure 7, the workspace is surrounded by three static and two dynamic obstacles. From both figures, it is found that the NNS model is capable of planning efficient paths for each agent by covering all the given target locations.

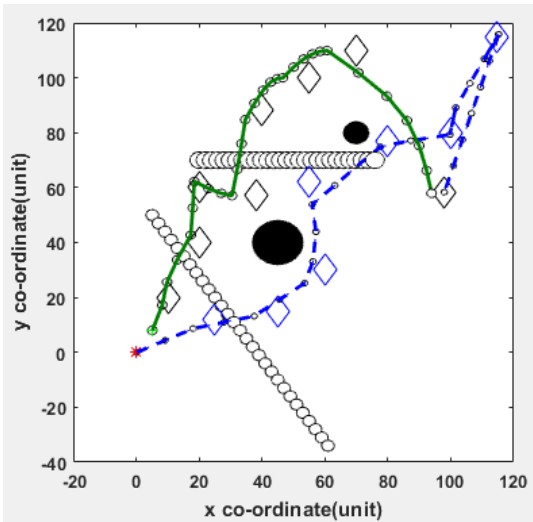

**Figure 4.** Path planning for two agent system with fifteen target locations.

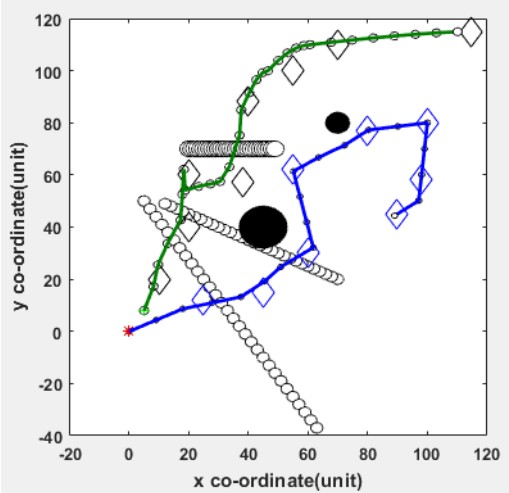

**Figure 5.** Path planning in dynamic environment with sixteen assigned target locations.

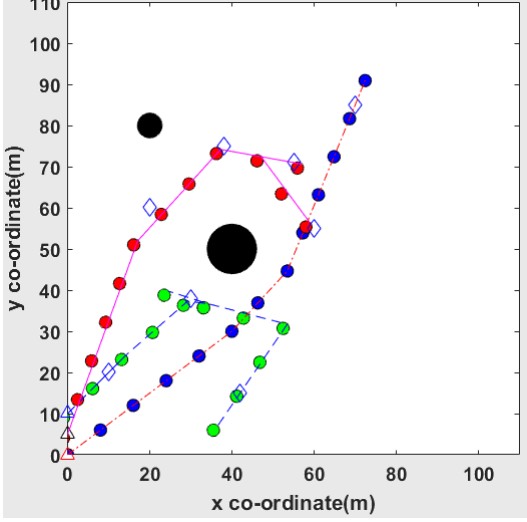

**Figure 6.** Three agents path planning and task assignment in static environment.

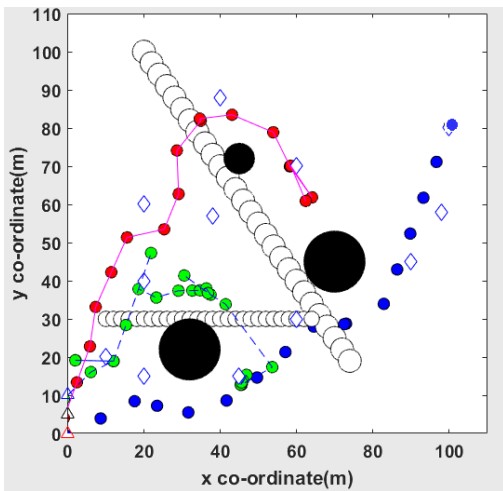

**Figure 7.** Three agents path planning and task assignment in complex environment.

Figure 8 shows the path planning of five agents in static environment. Figure 9 shows the path planning for three agents in complex environments. In both figures, agents complete their mission by visiting 20 target locations.

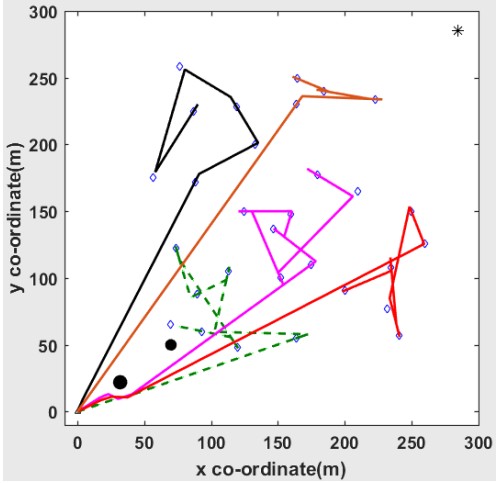

**Figure 8.** Five agents path planning and task assignment in static environment.

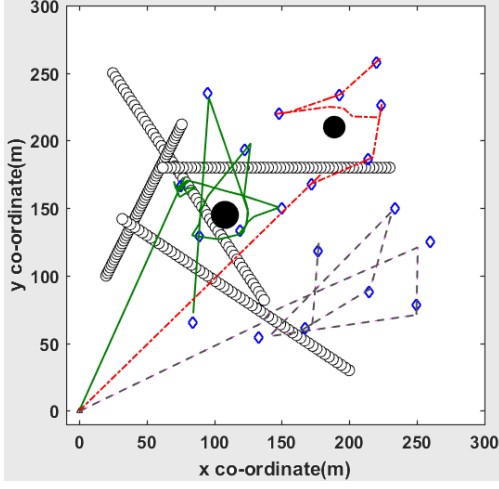

**Figure 9.** Three agents path planning and task assignment in complex environment.

From Figures 6–9, it is clear that the framework with the NNS model is applicable to the case where the number of agents and targets can be given randomly. Moreover, by using this framework agents can efficiently complete their mission in any types of environment.

### 5.2. Breakdown of some Agents

The NNS-based framework can efficiently tackle some uncertainties such as the breakdown of any agents on its way. Figures 10 and 11 show the path planning of 3 agents while visiting 20 target locations in static environments. Both figures show the situation where agent-1, agent-2 and agent-3 are assigned with 20 tasks. Number of tasks for agent-1, agent-2 and agent-3 are seven, seven and six, respectively.

In Figure 10 it is found that all three agents can successfully complete their assigned tasks, whereas in Figure 11 it is found that agent-3 is unable to continue its journey after visiting three target locations. However, the remaining two agents mutually complete the remaining tasks of agent-3. Finally, all the assigned tasks are completed by the agents.

Figure 12 shows a sudden change of situation, where two of four agents are broken down at some period of time. It describes the planning in a complex environment where the workspace is surrounded by two static and four dynamic obstacles. From Figure 12 it is found that agent-3 and agent-4 are not capable of doing their tasks after visiting one and four target locations, respectively. However, the other agents complete the rest of the tasks efficiently.

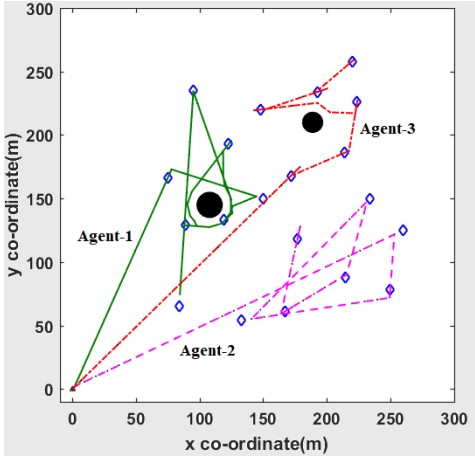

**Figure 10.** Three agents path planning and task assignment without any break down in static environment.

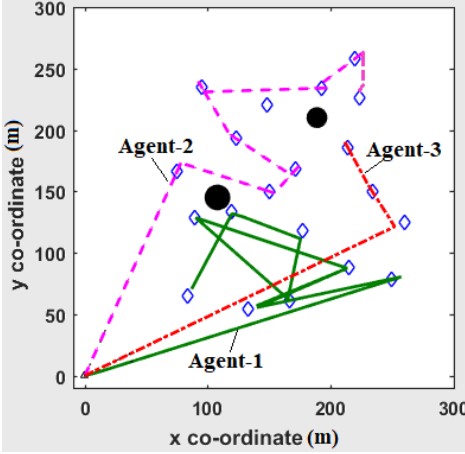

**Figure 11.** Three agents path planning and task assignment where one agent broken down on the way towards its targets.

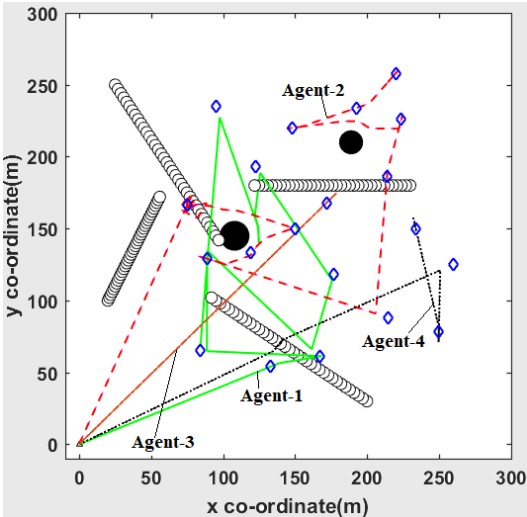

**Figure 12.** Four agents path planning and task assignment in dynamic environment where two agents broke down on their way to the targets.

### 5.3. Task Switching

The robustness of the NNS-based approach is that the agents are capable of switching their tasks. One agent can switch some of its tasks to another agent that has already finished its own assigned task.

Figures 13 and 14 show the situation where two agents plan their paths by completing their individual assigned tasks in a dynamic environment. The number of assigned tasks to agent-1 and agent-2 is 16 and 9, respectively. The number of assigned target locations to agent-1 is more than the number of assigned target locations to agent-2. Hence, agent-2 is able to finish all tasks before agent-1 finishes its given tasks. In this case, agent-1 switch some of its remaining tasks to agent-2. This will increase the effectiveness of the total framework. From Figure 14 it is found that task switching occurred between two agents.

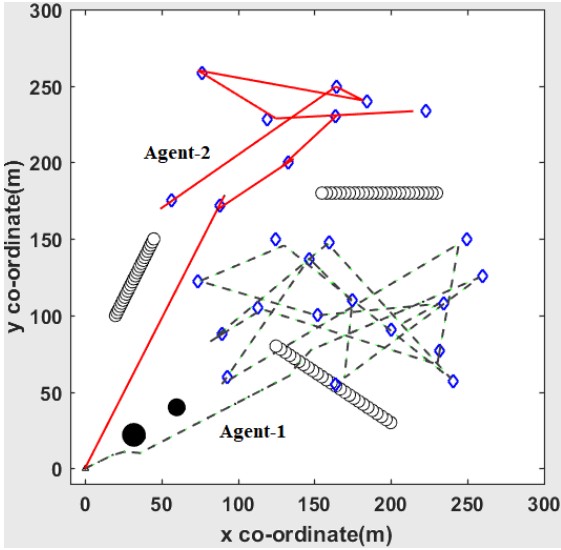

**Figure 13.** Two agents planning without any switching task.

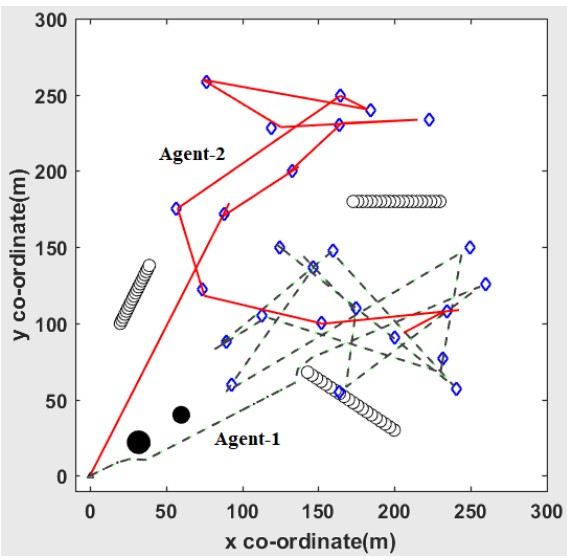

**Figure 14.** Two agents planning with task switching.

## 6. Conclusions

This paper focuses on the investigation and comparison of the NNS and CV model frameworks, for combining the task assignment with the path planning problem. These two frameworks are capable of cooperatively controlling a multi-agent system without any collisions. From the results, it is found that the NNS model shows better performance than CV model. Furthermore, NNS model is more time efficient than the CV model. However, it is only possible at the expense of path length. The NNS framework is robust. By using this model, agents are able to accomplish the given tasks by visiting all the target locations even in complex environments. This approach can tackle unpredictable situations such as the sudden breakdown of some agents. Furthermore, it can deal with the complicated situation where the number of agents is less than the number of targets. In addition, if required, the NNS model can increase the effectiveness of the total system by applying task switching phenomenon. That makes the NNS framework highly time-efficient. The analysis shows that it is very easy to implement and can be applied to any real-time environments.

**Author Contributions:** Conceptualization, S.B.; formal analysis, S.B.; investigation, S.B.; methodology, S.B.; supervision, S.G.A.; validation, S.B.; writing, original draft, S.B.; writing, review and editing, S.B., S.G.A. and M.A.G.

**Funding:** This research received no external funding.

**Acknowledgments:** The Research is supported by the Australian Government Training Program Scholarship. Our heartfelt thanks to The University of New South Wales at the Australian Defence Force academy, Canberra, Australia.

**Conflicts of Interest:** The authors declare no conflict of interest.

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
