# Peer review of "A Time-Efficient Co-Operative Path Planning Model Combined with Task Assignment for Multi-Agent Systems"

_robotics, doi:10.3390/robotics8020035_

Round 1
Reviewer 1 Report
This paper presents two frameworks of cooperative task assignment combined with path planning for multi-agent systems. Although the problem is interesting, the methods are extended straightforward from the authors' previous work [10] and [11]. I encourage the authors to hight the contribution of this work compared to their previous work.
From Table 1, what are the units of computational time and path length? Please explain more about calculating the computational time and path length.
In section 4.3, what is the explicit formula of the notation $Dp^{obs}$? From the definition (Line 319), $Dp^{obs}$ is a set of distance due to multiple obstacles and particles. However, in Line 320 and 321, it is used as a scalar. Moreover, it does not make sense that $Dp^{obs}<0$. This section should discuss in more detail what the collision avoidance strategy is.
To have a fair comparison in section 5.1, please consider the same setup (static and dynamics environments) for both approaches (CV based mode and NNS based mode). Moreover, please add some simulations show that the CV based framework cannot tackle the uncertainties such as the breakdown of any agents on its way?
Some minor issues:
Line 166: Caption of Figure 2 states that it is based on CV framework, but I do not see the CV model in the flow diagram.
Line 231: "Algorithm II and Algorithm III describes" ->"Algorithm II and Algorithm III describe".
Line 271: Please explain the notation $m$ and $n$. $m$ has been used in Line 142.
Line 307: "In multi agent planning" -> "In multi-agent planning".
Line 319: "Let, $Dp^{obs}$ is" -> "Let $Dp^{obs}$ be".
What is the difference between $Dp^{obs}$ in Line 319 and $D_p^{obs}$ in Line 217?
Author Response
the response is attached as a PDF file.

Reviewer 2 Report
1. In a nutshell, this paper essentially compares two of the author's previous works on PSO based multi-robot task cooperation and path planning. In this sense, the novelty of the paper is too weak for a journal paper. In fact, having read both the conference papers, the reviewer finds that most of the text and equations are reused from author's previous work. Some of the result figures are also similar, while others are relatively similar with less differences.
Biswas, S.; Anavatti, S. G.; Garratt, M. A. Nearest-neighbour based task allocation with multi-agent path 466 planning in dynamic environments, International Conference on Advance Mechatronics, Intelligent Manufacture 467 and Industrial Automation (ICAMIMIA), IEEE, October 12-14, Surabaya, Indonesia, 2017. 468
Biswas, S.; Anavatti, S. G.; Garratt, M. A. Particle swarm optimization based co-operative task assignment 469 and path planning for multi-agent system. IEEE Symposium Series on Computational Intelligence (IEEE SSCI), 470 November 27- December 1, Hawaii, USA, 2017, pp. 45- 50.
2. In Fig.1 and Fig.2 why have the authors used different types of arrows? Do they signify different things? These 2 figures are needlessly large, have different fonts, and can be drawn in simpler way. Caption of Fig.1 -> based on ???
3. Regarding Algorithms I, II, and III:
This is a terrible way to write pseudo-code. The code is unreadable. There is no alignment. Many variables are suddenly introduced and some parts are difficult to understand. For ex. line 244: "neighbor search (NNS)". Is this is a function called neighbor search which takes NNS as parameter? Lines 237-238 and 240-241 etc. can be combined into a single line with so much white space on the right side? Why have you split into separate lines with different line-numbers? In Algorithm III, for loop is started twice but ends only once. It is simply ugly. Please use Latex or flowchart for this and supplement with comments. Otherwise remove it altogether as it cannot be read or reused by anyone else.
4.How is 'k' or number of clusters automatically determined?
5. What is the space-complexity of the two algorithms.
6. In a sequential task assignment, how is it ensured that the appropriate robot is assigned the appropriate task. Ex. a robot is assigned a task which is nearest to its location.
7. As stated earlier, the simulation results look very similar to previous work. The setup is the same. It is currently difficult to understand what is new, as it is not explicitly stated anywhere. Most importantly, a comparison of the two algorithms with state-of-the-art techniques is missing.
8. The algorithm does not considers real-world constraints. (1) Robots have width and occupy space in the world and are not point-like entities as assumed bringing the robot too close to obstacles. (2) There is always some localization error due to sensor errors. (3) In dynamic scenarios, this localization errors keeps building up and robots need a SLAM algorithm to filter the uncertainties. (4) The paths are too zig-zag for real case scenarios and must be smooth. There are other real-world constraints which have not been taken into consideration while performing the simulations. Factors like 'safe distance threshold from obstacles' is not so difficult to implement. The assumption of types of sensors used is also missing. The reviewer has never seen a real-world scenario with three perfectly circular obstacles. Realistic scenarios include things like furniture, chairs, etc. and these are not difficult to include in simulations.
9. Lines 422-423:
"Hence, agent -2 is able to finish all tasks before agent-1finishes its given tasks. In this case, agent-1 switch some of its remaining tasks to agent-2."
How is this achieved? Is there an assumption of inter-agent communication?
10. It is not understood how is the solution 'optimal'? In other words, given certain conditions, how is it known that this is the best we can get? Paper lacks mathematical rigor in this regard.
11. Please check if all the variables used in the paper are properly defined.
Ex. see Eq.4. what is x_t^n etc.? Explicitly define everything or use a table of abbreviations in appendix.
What is the use of {} braces in Eq.4?
12. Why is the initial position of all the agents (0,0) and not something else. Random initialization should provide better insight into how algorithm performs. Similarly, why only 3 or 4 static obstacles?
13. Table 1: what is the unit of computational time? seconds? milliseconds? What is the unit of length? pixels? grid?
14. In all the result figures, please remove (unit) from x-coordinate(unit) etc. Everyone knows that there is some unit. What is the unit? Explicitly state it, or remove it.
15. In Fig.6, Fig.7, Fig.98, there is a strange blue rectangle dot on the top of text "x coordinate". Fig.6 when zoomed, the circle is crossing the bottom legend line around x=130 and the bottom line is broken. Please take journal paper seriously.
16. The reviewer strongly feels that there is no need to repeat what has already been published. Instead, authors should focus to bring novel contributions. They are strongly advised to do new experiments in realistic, and complex environments, with more and realistic setup of obstacles. The quality of figures, algorithm, table, etc. should be improved. Assumptions should be clearly stated. If authors need more time to revise the paper, they should contact the Editor. But major revision is very important to improve the quality of current work.
Author Response
The response is attached as a PDF file.

Round 2
Reviewer 2 Report
Authors have improved the paper well. I think that it can be accepted for publication after proof-reading.
Reviewer